# *GhENODL6* Isoforms from the Phytocyanin Gene Family Regulated Verticillium Wilt Resistance in Cotton

**DOI:** 10.3390/ijms23062913

**Published:** 2022-03-08

**Authors:** Man Zhang, Xingfen Wang, Jun Yang, Zhicheng Wang, Bin Chen, Xinyu Zhang, Dongmei Zhang, Zhengwen Sun, Jinhua Wu, Huifeng Ke, Liqiang Wu, Guiyin Zhang, Yan Zhang, Zhiying Ma

**Affiliations:** State Key Laboratory of North China Crop Improvement and Regulation, Key Laboratory for Crop Germplasm Resources of Hebei, Hebei Agricultural University, Baoding 071001, China; zhangman920601@163.com (M.Z.); cotton@hebau.edu.cn (X.W.); yang22181@163.com (J.Y.); zhichengwang775@163.com (Z.W.); chbhebau@163.com (B.C.); zxylhyh@163.com (X.Z.); zhangdongmei1108@163.com (D.Z.); sunzhengwen654@163.com (Z.S.); nxywjh@hebau.edu.cn (J.W.); kehuifeng123@126.com (H.K.); wuliqiang@hebau.edu.cn (L.W.); mhyzh@hebau.edu.cn (G.Z.)

**Keywords:** *Gossypium hirsutum*, *GhENODL6*, Verticillium wilt resistance, SA, ROS

## Abstract

Verticillium wilt (VW), a fungal disease caused by *Verticillium dahliae*, currently devastates cotton fiber yield and quality seriously, yet few resistance germplasm resources have been discovered in *Gossypium hirsutum*. The cotton variety Nongda601 with suitable VW resistance and high yield was developed in our lab, which supplied elite resources for discovering resistant genes. Early nodulin-like protein (ENODL) is mainly related to nodule formation, and its role in regulating defense response has been seldom studied. Here, 41 conserved ENODLs in *G. hirsutum* were identified and characterized, which could divide into four subgroups. We found that *GhENODL6* was upregulated under *V. dahliae* stress and hormonal signal and displayed higher transcript levels in resistant cottons than the susceptible. The *GhENODL6* was proved to positively regulate VW resistance via overexpression and gene silencing experiments. Overexpression of *GhENODL6* significantly enhanced the expressions of salicylic acid (SA) hormone-related transcription factors and pathogenicity-related (PR) protein genes, as well as hydrogen peroxide (H_2_O_2_) and SA contents, resulting in improved VW resistance in transgenic *Arabidopsis*. Correspondingly, in the *GhENODL6* silenced cotton, the expression levels of both phenylalanine ammonia lyase (PAL) and 4-coumarate-CoA ligase (4CL) genes significantly decreased, leading to the reduced SA content mediating by the phenylalanine ammonia lyase pathway. Taken together, *GhENODL6* played a crucial role in VW resistance by inducing SA signaling pathway and regulating the production of reactive oxygen species (ROS). These findings broaden our understanding of the biological roles of *GhENODL* and the molecular mechanisms underlying cotton disease resistance.

## 1. Introduction

Cotton is one of the most important textile crops worldwide and is regarded as the backbone of the economy of many developing countries [1]. Verticillium wilt (VW), caused by *V. dahliae*, a soil-borne hemi-biotrophic fungus, can lead to plant vascular discoloration and serious economic loss, and there is no effective fungicide available to control it. The yield damage of cotton caused by VW is approximately 10–35% each year [2]. For example, about 300 million hectares of cotton are subject to VW in China, and the economic loss is more than 12 billion Chinese Yuan per year [3,4]. Using genetic resistance is regarded as the most effective method in controlling VW in *G. hirsutum,* which accounts for over 90% of the yield in production; however, screening efforts globally examining thousands of cotton germplasm show few accessions displaying resistance [3,5]. Recently, we developed a resistant cultivar, Nongda601 (ND601), that could maintain a high yield in the field disease nursery [6], which supplied us with the possibility of further mining VW resistance genes. The high-quality genome of modern cultivated tetraploid cotton *G. hirsutum* released by our lab also promotes excavating functional genes and genome-wide analysis of gene families [5].

Plants have evolved sophisticated resistance mechanisms to defend against a diversity of biotic stresses [7]. In cotton, the defense mechanisms primarily depend on the defense structures, including thick cuticle, synthesis of phenolic compounds, reinforcement of cell wall structure, accumulation of reactive oxygen species (ROS), release of phytoalexins, and the hypersensitive response [8,9,10]. In addition, several signaling pathways participate in mediating the communications between the plant and pathogen interaction. Salicylic acid (SA) is one of the key defense-related hormones and may be induced by different pathogens and mediate gene expression in response to stresses [2,11]. ROS could regulate multiple signaling in a highly coordinated manner [12], including antioxidants, kinases, defense genes, the influx of Ca^2+^ ions, synthesis of plant hormones (such as SA, JA, and ET) [13], and directly involved in signal transduction and defense response, resulting in inhibiting fungal propagation [14]. It was reported that the Cys-rich repeat protein 1 (*CRR1*) protected chitinase 28 (*Chi28*) from cleavage by *V. dahliae* secretory Ser protease 1 (*VdSSEP1*) to facilitate defense against the fungal pathogen *V. dahliae* [15]. *GhLAC15* enhanced VW resistance via an increase in defense-induced lignification, arabinose, and xylose accumulation in the cotton cell wall [16]. *GbCYP86A1-1* positively regulated VW resistance by cell wall modification and activation of immune pathways [17]. Pectin lyase could improve cotton resistance by inducing cell apoptosis of *V. dahliae* [18]. To date, transgenic cotton with broad-spectrum disease resistance has not been released, although many attempts have been made, and the lack of powerful resistance genes has been a possible factor hindering the progress of disease-resistant transgenic cotton [19]. Thus, it is still necessary to deepen our understanding of the key factors and regulation mechanism of cotton resistance to VW and to mine more novel resistance genes for the future efforts of cotton improvement.

The accumulation of ROS is considered to be the earliest event caused by the plant-pathogen interaction, which could control and inhibit the growth of pathogens. ROS is also considered to be the crosslinking molecules in plant cell walls that block the entry of pathogens and act as local and systemic secondary messengers to trigger additional immune responses [20]. Hydrogen peroxide (H_2_O_2_) can promote the expression of the glutathione S-transferase (GST1) gene in adjacent cells, preliminarily demonstrating the signal transduction function of H_2_O_2_ in cells [21]. H_2_O_2_ also induces the increased activity of benzoate 2-hydroxylase that is required for the synthesis of SA necessary for the development of systemic acquired resistance (SAR) [22]. These findings suggested that H_2_O_2_ regulated and participated in SA signaling during signal transduction [23].

Several early nodulin-like proteins (ENODL) belonging to phytocyanin (PC) have been identified via VW-induced transcriptome analyses [24], which aroused our interest in the identification and analysis of this novel gene family. PCs can be divided into four main subclasses according to the identity of copper ligand residues: uclacyanins (UCs), stellacyanins (SCs), plantacyanins (PLCs), and early nodulin-like proteins (ENODLs) [25]. Miwa et al. analyzed PC genes in morning glory and found that they were involved in plant organ differentiation [26]. Cao et al. revealed diverse expression patterns of the PC genes in response to salt and drought stresses in maize and discovered that all nine detected maize PC genes were down-regulated under salt treatment, of which five PCs were down-regulated under drought treatment [27]. Xu et al. identified 30 putative PC genes in the *Phalaenopsis equestris* through comprehensive bioinformatics analysis, and expression analysis revealed that nine PC genes were highly expressed in flowers, stems, and roots [28]. Ma et al. identified the PC family in rice and appraised the expression profiles of *OsPC* genes at different development stages and under abiotic stresses [29]. However, in cotton, there is no report of PC family in response to *V. dahliae*, and the function of important PC family members are also obscure. Therefore, identifying the functional potential of such gene family and the key member with important functions is essential. Here, we identified 113 PC genes and 41 ENODL genes in cotton and determined the interactions between the ENODL subgroup and *V. dahliae*. Subsequently, we verified the function of a candidate gene, *GhENODL6*, in resistance to VW and revealed its molecular mechanisms of the disease resistance associated with the SA and ROS pathway.

## 2. Results

### 2.1. Identification, Sequence Features, and Phylogeny of the Cotton GhPCs Gene Family

To determine the *PC* genes in *G. hirsutum*, we performed a genome-wide prediction of *GhPC* genes by BLAST analysis of 38 *AtPC* against the *G. hirsutum* genome using the tBLASTn algorithm, SMART, and the Pfam (PF02298) [30]. A total of 113 *GhPC* genes from *G. hirsutum* were identified and numbered according to genome location. The open reading frames (ORFs) of *GhPC* genes ranged from 321 bp (*GhPC1*) to 2496 bp (*GhPC41*), the predicted molecular weight varied from 11.763 kDa (GhPC1) to 88.307 kDa (GhPC41), and the theoretical pI varied from 4.128 (*GhPC34*) to 10.573 (*GhPC12*) (Appendix A). To investigate the evolutionary relationships of the *GhPCs* proteins, sequences of the 113 GhPCs and *Arabidopsis* AtPCs proteins [31] were used to construct a neighbor-joining (NJ) phylogenetic tree. According to evolutionary relationships, 113 GhPCs were classified into five subgroups (subgroups a to c within class I and subgroups d and e within class II) (Figure 1), of which 69 GhPCs genes within class I had copper-binding sites with four conserved amino acid residues (His, Cys, His and Met/Gln), and the 69 proteins could be further divided into three subgroups, including 25 UCs (subgroup a), 32 SCs (subgroup b), and 12 PLCs (subgroup c), respectively. The remaining 44 GhPCs in class II included 41 ENODLs (subgroup d), and 3 unknown PCNL-containing proteins (subgroup e) were gathered due to the lack of key copper-binding amino acid residues. Moreover, 40 *GhENODLs* had putative N-glycosylation sites. Except for the PCNL domain, 31 ENODLs have presumed AG glycomodules in the PAST-rich region (rich in Pro, Ala, Ser, and Thr) (Appendix A).

### 2.2. GhPC Genes Have Distinct Expression Patterns upon V. dahliae Stress

Chromosome location analysis revealed that 41 *GhENODLs* were unevenly distributed on 16 chromosomes (Figure 2b), of which 35 *GhENODLs* contained signal peptides, and 22 were found to be GPI-anchored proteins, suggesting that these proteins might localize in the plasma membrane. In order to investigate the response of *GhPCs* to *V. dahliae* infection, we analyzed the transcriptional change of *GhPCs* based on our previous RNA-seq data [32]. Interestingly, we found that *GhPC* genes in different subgroups displayed diversity expression patterns, and only most *GhENODLs* in subgroup d showed obvious expression change under *V. dahliae* stress (Figure 2a). In detail, as many as 25 out of 41 *GhENODL* was upregulated at 2 h post inoculation (hpi), indicating that these genes potentially early regulated VW resistance (Figure 2a). Among the differentially expressed *GhENODLs*, *GhENODL6* displayed maximal fold-changes post inoculation based on both RNA-seq data and q-PCR results (Figure 2c); thus, *GhENODL6* was further used for functional study.

### 2.3. GhENODL6 Is a Plasma Membrane Protein

The *GhENODL6* (GhM_D06G0164; Ghir_D06G001340) has an open reading frame (ORF) of 420 bp encoding a 139 amino acid, the molecular weight of GhENODL6 predicted by the online tools ProtParam (http://web.expasy.org/protpara 12 March 2021) is 15.71 kDa, and the theoretical PI is 9.97. GhENODL6 contained 27-residue signal peptide (SP), an 86-residue PCNL domain and a transmembrane helix (12-34aa) (Figure 3a,b). To confirm its subcellular localization, the fusion protein of GhENODL6 with GFP under the constitutive CaMV 35 s promoter was successfully expressed in the epidermal cells of tobacco and onions, respectively, and we vividly observed the fluorescence signal both in cytoplasmic and membrane (Figure 3c and Appendix A), indicating that GhENODL6 was a secretory protein. Though it belonged to the PC family and was similar to the ENODL, *GhENODL6* lacked the copper-binding residues (Appendix A), suggesting that *GhENODL6* was an atypical copper ions member.

### 2.4. GhENODL6 Expression Is Upregulated in Response to V. dahliae and Hormonal Signal in Cotton

In order to investigate the *GhENODL6* expression in different tissues and VW-resistant cottons, we detected *GhENODL6* transcript levels and found that *GhENODL6* predominantly expressed in the root (Figure 4a), and *GhENODL6* could obviously induce upon *V. dahliae*, with higher expression level in resistant ND601 than in susceptible CCRI8 (Figure 4b). This trend was also evidenced in multiple cottons with different resistance [33], indicating that *GhENODL6* should positively regulate VW resistance (Figure 4c). Based on the hormonal signal treatment experiment, we found that *GhENODL6* dramatically upregulated expression upon foliar spray with JA and SA, especially under treatment by SA (Figure 4d), suggesting that *GhENODL6* is involved in SA and/or JA signaling pathway.

### 2.5. GhENODL6 Positively Regulated Verticillium Wilt Resistance in Plants

To confirm the function of *GhENODL6* in response to *V. dahliae*, we firstly overexpressed it in *Arabidopsis* and obtained T_3_ pure transgenic lines. Of the nine independent T_3_ transgenic lines, three stable overexpressing lines selected via RT-PCR and Western blot analysis were inoculated with *V. dahliae* (Figure 5a,b). Twenty days after inoculation by strong pathogenic strain Lx2-1 [34], typical symptoms of VW displayed evident in the infected wild-type (WT) plants but were much less pronounced in *GhENODL6* transgenic lines (Figure 5c). The statistical analysis showed that the average DI of transgenic lines was 32.0, significantly lower than that of WT, with a 45.8% decrease (Figure 5d). Additionally, the fungal biomass in the transgenic plants was markedly lower than in the WT (Figure 5e). Based on the above evidence, we concluded that overexpression of *GhENODL6* in *Arabidopsis* could improve the VW resistance of plants.

To further confirm the *GhENODL6* function in cotton, we successfully knocked down endogenous *GhENODL6* via virus-induced gene silencing (VIGS) that was successfully established in this research (Figure 6a). At 10 days post agroinfiltration, the transcript of *GhENODL6* was significantly suppressed, decreasing 75% of expression level in silenced plants (Figure 6b). The silenced and mock cotton displayed a significant difference in disease symptoms at 20 days post inoculation (dpi) under Lx2-1 stress, as indicated in DI and disease-grade statistical results (Figure 6c–e). Moreover, the fungal biomass in silenced plants was significantly higher than that in mock plants (Figure 6f,g). The VIGS experiment suggested that suppressing *GhENODL6* could make cotton disease reaction type from tolerant to susceptible. Combining the results of transgenic *Arabidopsis* and silenced cotton, we proved that *GhENODL6* positively modulated plant VW resistance, resulting in disease resistance from susceptible to tolerant grade.

### 2.6. GhENODL6 Regulates Defense Genes Involving in SA Signaling

To identify the molecular basis of *GhENODL6* in response to the pathogen, RNA sequencing (RNA-seq) was carried out to analyze the transcriptome profile of transgenic plant leaves in the absence of *V. dahliae*. Comparative transcriptome analysis between transgenic plants and WT, a total of 4235 differential expression genes (*q* < 0.05) were identified, including 2170 upregulated and 2065 down-regulated genes. Gene ontology (GO) analysis showed that a subset of genes involved in defense response to fungus, response to immune response, secondary metabolite biosynthetic process, response to oxidative stress, response to antibiotic, glucosinolate biosynthetic process, response to SA, response to ROS, and defense response by callose deposition were enriched (Figure 7a and Appendix A). We further compared the differentially expressed genes (DEG) between transgenic plants and WT at 24 hpi when *GhENODL* peaked expressing under *V. dahliae* stress. Interestingly, the number of DEGs showed remarkable differences. In detail, 4235 differential genes were found in the *GhENODL6-4* transgenic line, and these differential genes were mainly related to the response to a variety of abiotic factors, as well as carbohydrate catabolic process, defense response, incompatible interaction, response to SA and ROS (Appendix A). In contrast, 783 differential genes were found in WT, in addition to the response to abiotic factors, genes mainly involved in response to extracellular stimulus defense response to fungus (Appendix A). Compared with the WT, more resistance genes were activated in *GhENODL* transgenic plants, and KEGG revealed phenylpropanoid biosynthesis was enriched (Figure 7b). Given the fact that the phenylalanine ammonia lyase pathway is the main synthesis pathway of SA in cotton [35] and the results of qPCR in Figure 4e and the RNA-seq data, we deduced that *GhENODL6 is* probably involved in SA signaling. Thus, the SA pathway-regulated genes, *EDS1, PR1, PR2, PR5,* and *PR10*, were detected via qPCR (Figure 8a–e). Compared to the WT, the transcript levels of the above genes were significantly upregulated in transgenic lines, indicating that overexpression *GhENODL6* could influence SA-mediated defense response. In contrast, the expression of those genes was observably hindered after suppressing *GhENODL6* in silenced cotton (Figure 8g–k). The SA content also exhibited consistent trends, with increasing in the overexpression *Arabidopsis* and decreasing in the silenced cotton, respectively (Figure 8f,l). The expression levels of *PAL*, *4CL* involved in the phenylalanine ammonia lyase pathway were also decreased in the silenced plant, and the activity of PAL was significantly lower than the control group (Figure 8m–r). At the same time, the silenced plants and WT were treated by 1 mM SA via exogenous spraying, and we found that the silenced plants displayed improving disease resistance, with significantly lower DI than the WT (Figure 8s,t), suggesting that exogenous SA compensated the resistance of GhENODL6-silenced plants. Taken together, the above results indicated that *GhENODL6* defense response against *V. dahliae* was via activating the SA pathway.

### 2.7. GhENODL6 Contributed to the Production of ROS

It was well known that there was an important relationship between the production of ROS and the defense response of plants against pathogens. In order to determine whether *GhENODL6* is involved in the production of ROS, *Arabidopsis* and VIGS cotton plants were inoculated with *V. dahliae* and stained with 2,7′-dichlorodihydrofluorescein diacetate (H_2_DCFDA) and 3-3′-diaminobenzidine (DAB), and the results demonstrated that the ROS levels were remarkably higher in mock cotton leaves than in silenced cotton leaves (Figure 9a,b), indicating that *GhENODL6* was involved in the production of ROS in the immune response of plants to *V. dahliae*. We also determined H_2_O_2_ and superoxide dismutase (SOD) both in transgenic *Arabidopsis* and silenced cotton under *V. dahliae* stress. We found that both H_2_O_2_ content and SOD activity in the transgenic lines were significantly higher than in WT, while the silenced cotton displayed decreased trends compared to the mock cotton plants (Figure 9c–f). These results suggested that *GhENODL6* was related to ROS production during plant immune responses to *V. dahliae*.

## 3. Discussion

PCs are ancient blue copper-binding proteins in plants. They act as electron transporters and bind to a single type I copper atom. PCs play important roles in plant development and stress resistance and can be divided into four subfamilies: UCs, SCs, PLCs, and ENODLs, and all PCs have a PCNL domain [27]. The structure of ENODL is similar to that of the other three subfamilies, while ENODL lacks amino acid residues that bind to copper. The vast majority of ENODL are chimeric arabinogalactosins (AGPs) [28,36] and relate nodule development [37], pollen germination [38] and osmotic tolerance [39]. Until now, there is no report of *ENODLs* involved in disease resistance. This research was the first to link cotton *ENODL* with VW resistance, and this will broaden our knowledge toward *ENODLs’* biologically specific function in crops.

In this study, the *ENODLs* sub-family was identified responding more to *V. dahliae* than the other three subgroups (SCs, UCs, and PLCs). There was rare information about the functions of *ENODLs* in plants. Here, we found that the *GhENODL6* was significantly induced by *V. dahliae* from 2 hpi, suggesting its potential function on early regulating VW resistance. We also found that *GhENODL6* expression peaked at 24 hpi, which guided us to sample the materials at 24 hpi by *V. dahliae* for comparing the DEGs between transgenic and WT plants, which should reflect the relatively big expression of the defense genes between transgenic and WT plants. *GhENODL6* was proved to positively regulate VW resistance and displayed transcriptome differences between resistant and susceptible cottons. Functional studies firstly demonstrated that *GhENODL6* could improve plant VW resistance via the SA signal pathway and ROS production, which could make the plant disease reaction susceptible to tolerance. These results indicated that *GhENODL6* was a functionally novel gene in defense against VW resistance.

SA is a key defense-related hormone in plants, which plays an important role in SAR and broad-spectrum durable disease resistance [40,41]. Among the reported plant defense hormones, the roles of SA, JA, and ethylene (Eth) have been established, and these hormones play important roles in biological stress signaling pathways, including ROS and MAPK kinases [8]. In cotton, most SA biosynthesis relies on the phenylalanine ammonia lyase pathway [35]. In this study, according to qPCR results and RNA-seq data, we inferred that *GhENODL6* could activate the phenylpropanoid pathway and promote SA synthesis. Both hypersensitive reaction (HR) and SAR were associated with increased expression of a wide range of defense genes, including those encoding various types of pathogen-related (PR) proteins. The induction of the PR gene is closely related to the occurrence of disease resistance, so the PR gene is a common molecular marker of defense response [34]. Meanwhile, as an important marker gene in response to SA, PR genes play an essential role in plant immune response [42]. In this study, our findings demonstrated that *GhENODL6* responded quickly to SA signaling and maintained a high expression level, suggesting that *GhENODL6* was related to the SA signaling pathway. The activation of SA signals in stressed plants can stimulate the expression of downstream disease resistance genes and provide protection [43]. There are many genes related to the SA signaling pathway, including NPR1, PR1, PR2, WRKYs, and GSTF6 [44]. To verify this, we tested the expression level of SA-related marker genes in overexpressing *GhENODL6* transgenic *Arabidopsis* and silenced cotton. The expression of SA signaling pathway-related genes (*EDS1*, *PR1,* and *PR2*) and the increase in SA content confirmed our hypothesis. In order to further clarify the connection between *GhENODL6* and SA signal, we treated silenced cotton and mock plants with exogenous SA and observed that exogenous SA significantly increased the resistance of silenced plants to *V. dahliae*. These findings demonstrated that *GhENODL6* participated in the SA signaling pathway and functioned in VW resistance.

Plants infected by pathogenic bacteria and fungi would produce ROS, of which H_2_O_2_ played multiple roles in the interactions between the host and pathogens and was considered to be an antibacterial agent in the defense response of plants [45,46]. It also further participated in different signaling pathways related to defense mechanisms, such as triggering allergic reactions, accumulation of phytoantitoxins, and activation of many other defense response genes [47]. In order to overcome oxidative damage, plants have an antioxidant defense system composed of a variety of enzymes, such as SOD, which can remove, neutralize and scavenge active oxygen by converting superoxide anion radical (O2.-) into H_2_O_2_ and O_2_ [48]. In this study, the H_2_O_2_ content and endogenous SOD activity of *GhENODL6* transgenic *Arabidopsis* and *GhENODL6* silenced cotton after *V. dahliae* infection were determined. The H_2_O_2_ content and SOD activity of transgenic *Arabidopsis* were substantially higher than WT; as expected, silencing *GhENODL6* reduced H_2_O_2_ accumulation and endogenous SOD enzyme activity. Thus, we deduced that the role of *GhENODL6* in plant fungal resistance might also be due to the accumulation of H_2_O_2_ regulated by SOD activity.

This study is the first to discuss the relationship between ENODL and disease resistance. The function of *GhENODL6* was demonstrated through gene transcription, gene overexpression, and gene silencing experiments, highlighting the *GhENODL6* novel role in VW resistance. The molecular network of *GhENODL6* is still unclear at present. Based on bioinformatics prediction and yeast two-hybrid assay, we preliminary obtained several candidates interactors, including 14-3-3 protein 6, ethylene-responsive transcription factor RAP2-7, and CBL-interacting serine/threonine-protein kinase 3, which inspired us to study the mechanism underlying *GhENODL6* responding to disease resistance in the future.

## 4. Methods

### 4.1. Plant Growth and V. dahliae Culture

Seeds of *G. hirsutum* cv. CCRI8 and cv. ND601 (G100729) were delinted with H_2_SO_4_ (98%) and rinsed in water, then soaked in distilled water for 1 day and then germinated on wet gauzes for another day at 25 °C. Germinant seeds were transferred to pots containing vermiculite in a controlled environment (25 °C, 16 h/8 h (day/night), and 75% relative humidity). *Arabidopsis* were grown in growth chamber with a climate (23 °C, 16 h/8 h (day/night), and 60–70% relative humidity) [49]. The defoliating *V. dahliae* isolate (Lx2-1) was grown on potato dextrose agar medium for 7 days and then cultured in Czapek’s broth at 150 rpm, 25 °C for 1 week. When inoculating spores, adjust the concentration to 1 × 10^7^ conidia/mL with deionized water. For specific methods, refer to Yang et al. that pipetting conidia to the roots using injection method [50].

### 4.2. Identification of GhPCs and Bioinformatics Analysis

We first used the plastocyanin-like domain (PCNL) from Pfam (PF02298) (http://pfam.sanger.ac.uk/ 12 March 2021) to screen the protein databases in *Arabidopsis* and cotton. The sequences were then confirmed by the databases NCBI CD-Search. Subsequently. The N-terminal signal peptide (SP) of all proteins was examined using SignalP 3.0 [51]. Big-PI Plant Predictor was used to predict the glycosylphosphatidylinositol (GPI) modification site [52]. Moreover, we also used NetNGlyc 1.0 Server to predict the N-glycosylation sites [53]. Putative arabinogalactan (AG) glycomodules were predicted mainly following the previously described criteria [54].

### 4.3. Multiple Sequence Alignment of PCNLs and Phylogenetic Analysis

The Clustal W [55] program was used to perform multiple sequence alignments of all PCs protein sequences in *Arabidopsis* and cotton. The phylogenetic tree was constructed using the neighbor-joining (NJ) method of MEGA 7.0 software, and the bootstrap replications were set to 1000. The gene structures, chromosomal distribution, conserved motifs were analyzed referred to Liu et al. [56].

### 4.4. Expression Analysis of GhPC Genes

The transcriptome database of ND601 infected by *V. dahliae* was obtained from our laboratory [32]. A heatmap of PC gene expression clusters was constructed using TBtools software, and the expression values of PC genes were presented as fragments per kilobase of exon model per million mapped reads (FPKM). Finally, log_2_ (FPKM) values after averaging two replicates were displayed.

### 4.5. Gene Cloning and Bioinformatics Analysis

The root tissue of cotton seedlings was ground with liquid nitrogen, and total RNA and DNA were extracted using an EASYspi plant RNA rapid extraction kit (AidLab, Beijing, China) and EasyPure^®^ Plant Genomic DNA Kit (TransGen, Beijing, China), respectively. cDNA was obtained by reverse transcription using Prime Script II 1st Strand cDNA Synthesis Kit (Takara, Dalian, Japan). We designed a pair of primers for PCR amplification based on the homologous sequence of upland cotton TM-1 in Cotton FGD (https://cottonfgd.org 12 March 2021). The primers are shown in Appendix A. The sequence of *GhENODL6* was analyzed by Blast in the NCBI database (http://www.ncbi.nlm.nih.gov/ 12 March 2021), and the functional domain of the protein encoded by the gene was analyzed at the same time. The protein signal peptide was predicted by online software SignalP 4.0 (http://www.cbs.dtu.dk/services/SignalP 12 March 2021). The protein transmembrane domain was analyzed by TMpred (http://www.ch.embnet.org/software/TMPRED_form.html 12 March 2021).

### 4.6. Real-Time PCR and Expression Analysis

Cotton seedlings were treated under multiple conditions. Cotton seedlings were cultivated in nutritious soil and were treated with hormone spraying at the 4 leaf stage. Ethephon and SA were diluted to 5 and 10 mM, respectively, and sprayed evenly on the leaves of cotton seedlings and moisturized with a transparent plastic cover. The leaves of cotton seedlings were taken at 6, 12, 24, 36, and 48 h, respectively, and stored at −80 °C. Subsequently, cotton seedlings were grown in nutrient medium and inoculated with the root dip method. The specific operation is as mentioned before [57]. After inoculation, the cotton seedlings were cultured under the conditions of 25 °C, 14 h light/10 h darkness, and 70% relative humidity. The root tissues of cotton seedlings were taken at 6, 12, 24, 36, and 48 h after inoculation and stored at −80 °C. The above tissues were sampled at each time point and repeated three times. Distilled water treatment was used as a control. Total RNA was extracted from roots, stems, and leaves for tissue expression analysis.

The real-time PCR reaction system (20 μL) consists of 1 μL cDNA template (<100 ng), 10 μL SYBR^®^ Premix Ex TaqTM (TaKaRa), forward and backward primers 0.8 μL (10 μmol/L), and 7.4 μL sterilized double distilled water). The reaction was carried out in CFX96 real-time PCR detection system (Bio-Rad), and the program was set up as follows: 95 °C, 15 s, 95 °C, 10 s, 58 °C, 10 s, 72 °C 15 s, 40 cycles. Poly ubiquitin 14 (UBQ14) was used as the internal reference. Each reaction was repeated three times. The primers used are shown in Appendix A. The comparative CT (2^−ΔΔCT^) method was used for relative quantitative analysis. GraphPad Prism^®^ 6 software was used for statistical analysis and drawing [58].

### 4.7. Subcellular Location

In order to detect the subcellular localization of GhENODL6, we used tobacco for transient expression. The target gene was cloned into pMDC43-GFP vector by gateway method [59] and verified by sequencing. It was transferred into *Agrobacterium tumefaciens* strain GV3101 by the freeze-thaw method [60]. The transient transformation of tobacco leaf epidermis cells was described by predecessors [61]. The leaves were observed with a confocal laser scanning microscope (Fluo View FV1000; Olympus, Tokyo, Japan) about 2–3 days later.

### 4.8. Generation and Analysis of Transgenic Arabidopsis

The coding sequence of *GhENODL6* was amplified using PCR from the cDNA of upland cotton ND601 and cloned into a pGWB414 vector stored in our laboratory. The recombinant plasmid was transformed into *Arabidopsis thaliana* plants by the floral dip method, and transgenic plants were selected on MS medium containing 50 mg/mL kanamycin [62,63]. Gene-specific primers were used to isolate homozygous plants and confirm transcription status and normal protein expression. T_3_ transgenic pure lines were obtained for subsequent experiments.

The resistance of *Arabidopsis thaliana* transformed by *GhENODL6* to *V. dahliae* was determined by the root dip method [64]. The 20-day-old *Arabidopsis thaliana* was uprooted, soaked in a suspension containing 1 × 10^7^ conidia per milliliter for 5 min, and then replanted in the soil. The disease index (DI) is graded according to a previous report [65]. The resistance mechanism of *GhENODL6* in transgenic *Arabidopsis thaliana* was analyzed by the determination of physiological indicators after inoculation. In addition, qRT-PCR was used to compare the expression of the internal defense-related genes of *GhENODL6* transgenic *Arabidopsis* with WT, with the *AtEF-1α* gene as internal standard [40,66].

### 4.9. VIGS Experiment in Cotton

Design primers *GhENODL6*-V-F and *GhENODL6*-V-R (Appendix A), containing restriction sites *Eco*RI and *Kpn*I for *GhENODL6* ORF sequence by PCR amplification and then ligated into VIGS vector *pTRV2* [67]. The recombinant plasmid was transformed into *Agrobacterium* GV3101 competent cells by the electric shock method [68]. In the cotton seedling stage, about 7–10 days (cotyledons fully unfolded, true leaves did not grow), with chloroplasts alterados1 gene (*CLA1*) as a marker gene, injected with *Agrobacterium tumefaciens* culture. The cotton VIGS was carried out according to the method previously described [69]. About 15 days after the VIGS procedure, detecting the silencing efficiency and then the plants were inoculated with *V. dahliae*. The experiments were carried out using at least 30 plants per treatment and repeated three times. The DI was calculated as described previously [64].

### 4.10. ROS Dyeing

ROS staining adopts a reactive oxygen detection kit (aladdin). Dilute DCFH-DA 1:1000 with PBS buffer to make the final concentration 10 uM. DCFH-DA completely immersed the cotton leaves and reacted at 28 °C for 20 min [70]. Before observation, the leaves were washed with double distilled water at least three times to remove the reagents on the surface and then observed the staining signal with a laser confocal microscope [71]. In order to observe the accumulation of H_2_O_2_, using the plant hydrogen peroxide DAB staining kit (servicebio), fresh cotton leaves were taken, incubated in 1 mg/mL, pH 3.8 DAB solution for 8 h, and then decolorized in 95% ethanol [72].

### 4.11. Determination of H_2_O_2_ and SOD Content

*V. dahliae* was inoculated into 4-week-old *Arabidopsis thaliana* plants by the method of Zhang et al. [24]. Then the leaves of the transgenic plant were taken at 6, 12, 24, 36, and 48 h, respectively. The activity of H_2_O_2_ and SOD was determined according to the kit’s instructions (Jiancheng Biotech Inc., Nanjing, China). The protein content was determined using the BCA protein detection kit (Sangon Biotech, Shanghai, China).

### 4.12. Determination of SA Content

In order to determine the concentration of endogenous SA in *Arabidopsis thaliana* and cotton, 100 mg leaf samples were extracted with methanol. The contents of SA were determined by ESI-HPLC-MS/MS and the standard curve method [34].

### 4.13. Transcriptome Sequencing

*GhENODL6* transgenic *Arabidopsis* plants and WT were grown in vermiculite for four weeks; the whole plants were collected treatment with Lx2-1 or water. Total RNA was isolated, and the libraries for sequencing were constructed. Sequencing was performed on the Illumina HiSeq2000 platform. GO analysis of the differentially expressed genes in the biological process was conducted using the AgriGO software [73]. GO terms with a corrected *p*-value less than 0.05 were considered significantly enriched by differentially expressed genes. GO is divided into three independent ontologies, which are biological process (BP), molecular function (MF), and cellular component (CC). The gene ontology database we used was Intepro, which was made up of CATH-Gene3D, CDD, MobiDB, HAMAP, PANTHER, PIRSF, ProDom, PROSITE, SFLD, SMART, SUPERFAMILY, TIGRFAMS, and Pfam. The Intepro was a common database for all the plants. The background was constituted by the whole annotated gene sequence of *Arabidopsis,* and the output of enrichment needed padj <0.05. All samples contained three biological repeats. Other relevant analyses were referred to previous results [17].

### 4.14. Primers for Gene Cloning, Vector Construction and Expression Analysis

All primers used in this study are listed in Appendix A. The primers used in this study were designed via Primer Premier 5 software and primer blast analysis.

### 4.15. Statistical Analysis

All experiments were performed at least three times for each determination. Statistical analysis was performed using Graph Pad Prism^®^ 6 software (Graph Pad, San Diego, CA, USA). Data were evaluated by analysis of variance (ANOVA method) followed by Dunn’s multiple comparisons test. The *p*-value ≤ 0.05 is considered to be statistically significant.

## Figures and Tables

**Figure 1 ijms-23-02913-f001:**
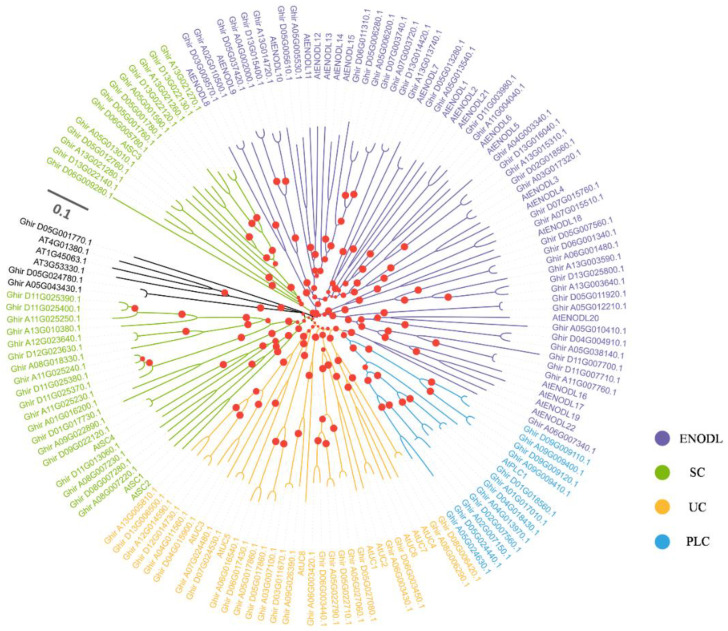
Phylogenetic analysis of PCs from two different plants. The phylogenetic tree was generated from the alignment result of the full-length amino acid sequences by the neighbor-joining (NJ) method. All PCs members, together with homologs of *A. thaliana*, were classified into four distinct clades shown in different colors.

**Figure 2 ijms-23-02913-f002:**
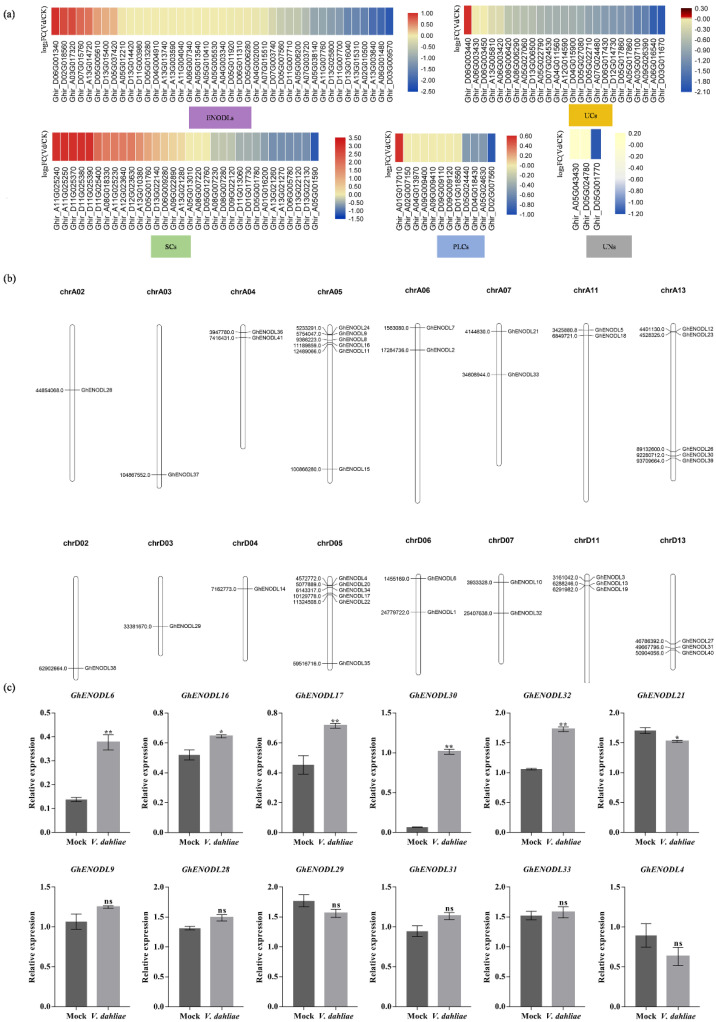
Expression analysis and chromosomal localization of ENODLs. (**a**) Expression patterns of ENODL genes in response to *V. dahliae* (using VDlog_2_ (FPKM)/CKlog_2_ (FPKM) to draw the graph) red: positive regulation, green: negative regulation. (**b**) Chromosomal distribution of *GhENODLs*. (**c**) Verification of the expression of *GhENODL* genes under the stress of *V. dahlia* (*, *p* < 0.05 **, *p* < 0.01). ns, not significant.

**Figure 3 ijms-23-02913-f003:**
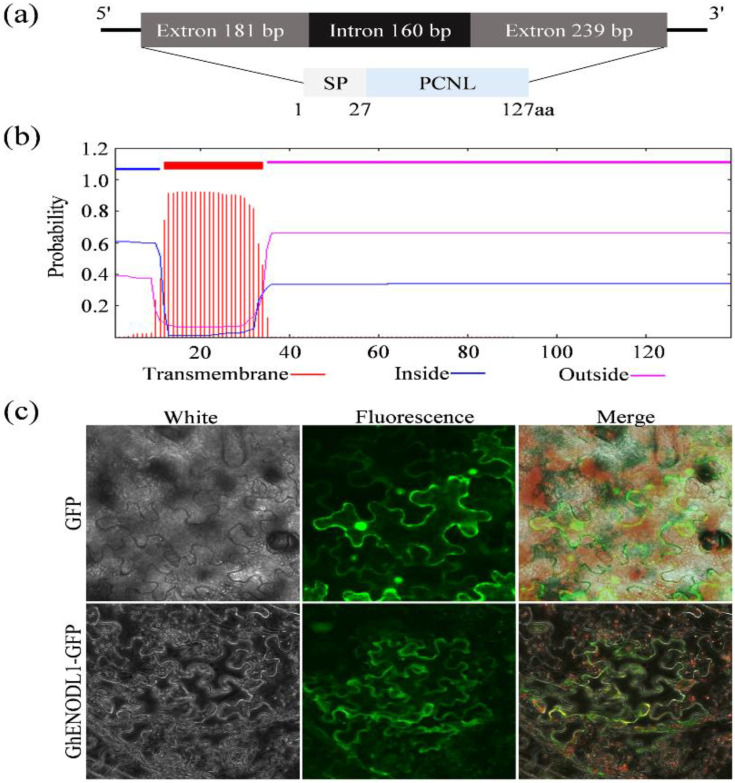
Characterization of *GhENODL6*. (**a**) Schematic structure of *GhENODL6*. (**b**) Prediction of the transmembrane helix in *GhENODL6* protein. (**c**) Subcellular localization of GFP alone or GhENODL6-GFP fusion in tobacco leaves transiently transformed by Agrobacterium infiltration. The green fluorescence was monitored using a confocal laser scanning microscope.

**Figure 4 ijms-23-02913-f004:**
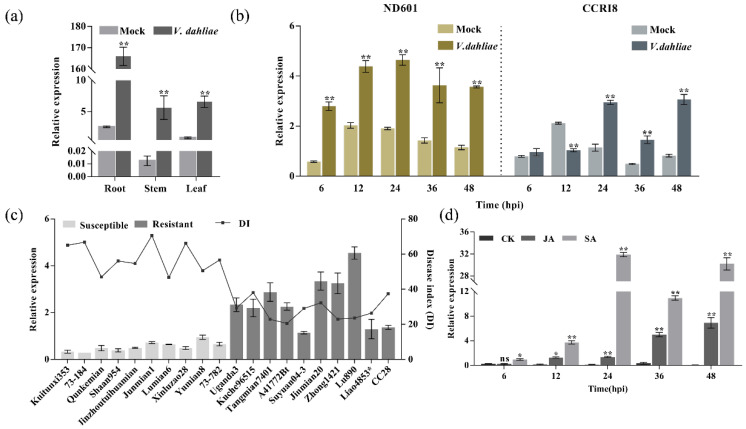
Expression analysis of *GhENODL6.* (**a**) Expression analysis of *GhENODL6* in different organs of cotton plant under Verticillium wilt stress. (**b**) Expression analysis of *GhENODL6* in ND601 and CCRI8 roots under VW stress. (**c**) Expression of *GhENODL6* in ten resistant and ten susceptible upland cotton cultivars upon infection with *V. dahliae*. (**d**) Expression analysis of *GhENODL6* with the treatment of water, JA, and SA, respectively. All experiments were repeated at least three times. Values are means with standard deviation (SD) (*n* = 3 biological replicates). Error bars represent the SD of three biological replicates. Asterisks indicate statistically significant differences according to Student’s *t*-test (two-tailed) (*, *p* < 0.05 **, *p* < 0.01). All experiments were repeated at least three times. Vd, *V. dahliae*; CK, control; ns, not significant.

**Figure 5 ijms-23-02913-f005:**
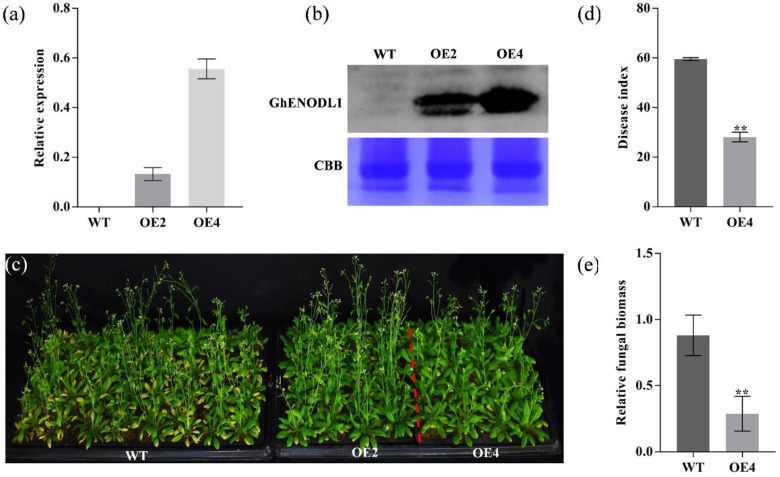
Overexpression of *GhENODL6* in *Arabidopsis* improved resistance to *V. dahliae*. (**a**) Relative expression levels of *GhENODL6* in different *Arabidopsis* strains. (**b**) Western blot result of *GhENODL6* in transgenic lines and WT. The membranes were stained with Coomassie brilliant blue (CBB) as loading control. (**c**) Disease manifestations and diseased plants of wild-type and two homozygous transgenic plants (OE2, OE4) at 25 dpi. (**d**) Disease index of different strains at 25 dpi. (**e**) Relative expression levels of plant fungus biomass at 10 dpi. Error bars represent the SD of three biological replicates. Asterisks indicate statistically significant differences by Student’s *t*-test (two-tailed) (**, *p* < 0.01). All experiments were repeated at least three times.

**Figure 6 ijms-23-02913-f006:**
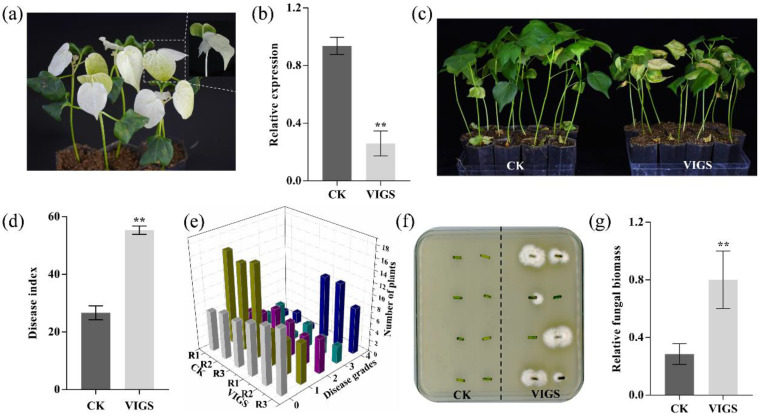
Silence detection of *GhENODL6.* (**a**) The *CLA1* gene was used as a positive control with an albino phenotype on leave after VIGS in cotton. (**b**) Relative expression levels of *GhENODL6* in TRV.*00* and TRV.*GhENODL6* plants. (**c**–**e**) The VW and statistics of the seedling number with different disease-grade symptoms of silenced and TRV.*00* plants at 25 dpi and disease index of seedlings at 25 dpi. (**f**) *V. dahliae* isolation from the infected stems. (**g**) Plant fungal biomass and relative expression levels. Error bars indicate the SE of three biological replicates. Asterisks indicate statistically significant differences as determined by Dunnett’s multiple comparisons test (**, *p* < 0.01).

**Figure 7 ijms-23-02913-f007:**
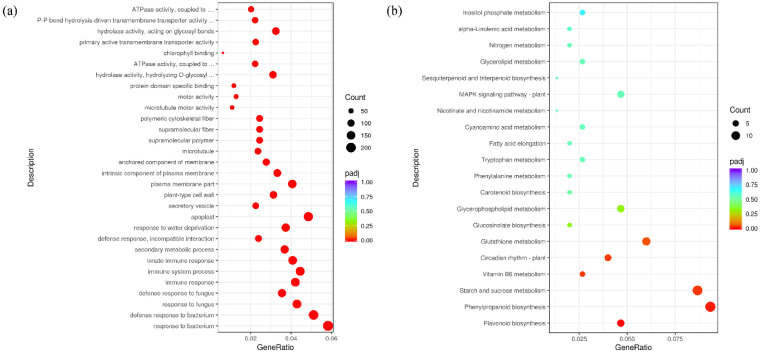
RNA-seq reveals that *GhENODL6* overexpression affects immunity-related pathways of *Arabidopsis*. (**a**) GO enrichment analysis of the differentially expressed genes in plants of *GhENODL6-4* transgenic line and the WT control. Padj. Corrected *p*-value after multiple hypothesis testing. GeneRatio: ratio of the number of differential genes annotated to the GO number to the total number of differential genes. (**b**) KEGG enrichment analysis of the difference in plants of *GhENODL6-4* transgenic line and the WT control after challenged with *V. dahliae* for 24 h. KEGG terms were separately shown by enrichment analysis. *p*-value of 0.05 adjusted by multiple hypothesis testing.

**Figure 8 ijms-23-02913-f008:**
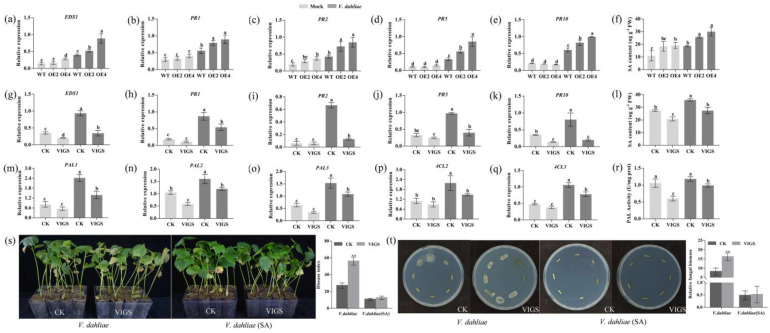
Detection of SA pathway genes. (**a**–**e**) qRT-PCR analysis of the expression of SA pathway genes in wild-type, transgenic *Arabidopsis* line. (**g**–**k**) qRT-PCR analysis of the expression of SA pathways in controls and silenced cotton plants. (**f**,**l**) Measurement of SA content in transgenic *Arabidopsis* and *GhENODL6*-silenced cotton. The leaves of *Arabidopsis* and cotton inoculated by *V. dahliae* for 2 dpi were used for SA. (**m**–**q**) qRT-PCR analysis of the expression of PAL pathways in controls and silenced cotton plants. (**r**) PAL active identification. (**s**) Disease symptoms induced with *V. dahliae* on the *GhENODL6-*silenced plant and assessment of disease Index (DI). Photos were taken at 25 dpi. (**t**) *V. dahliae* isolation from the infected stems and plant fungal biomass and relative expression levels. Values are means with SD (*n* = 3 biological replicates). Error bars represent the SD of three biological replicates (*p* < 0.05). All experiments were repeated at least three times. Lowercase letters represent statistically significant differences (*p* < 0.05) according to Tukey’s HSD test. All experiments were repeated at least three times (**, *p* < 0.01).

**Figure 9 ijms-23-02913-f009:**
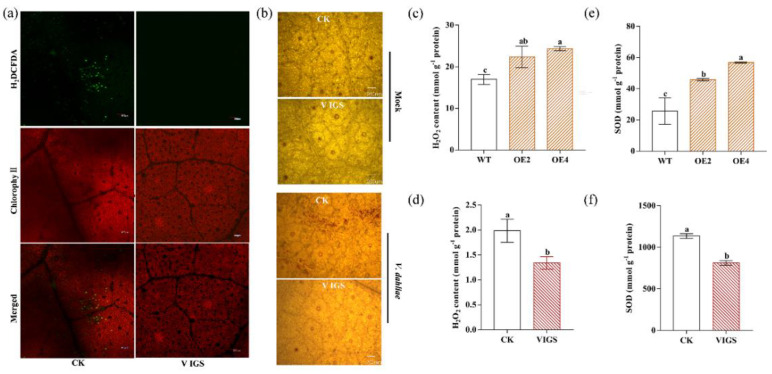
Determination of H_2_O_2_ content and activities at 2 dpi. (**a**) Detection of ROS levels in cotton leaves by H_2_DCFDA staining. Then 10 mM H_2_DCFDA ROS—specific fluorescent indicator was used for the staining. Bars = 100 um. (**b**) H_2_O_2_ visualization in cotton leaves by staining with DAB, showing more H_2_O_2_ was produced in the *TRV*.*00* than the silent plants. Bars = 200 um. (**c**–**f**) Measurements of the contents of H_2_O_2_ and SOD in cotton and *Arabidopsis* leaves at 2 dpi. Values are means with SD (*n* = 3 biological replicates). Error bars represent the SD of three biological replicates. Lowercase letters represent statistically significant differences (*p* < 0.05) according to Tukey’s HSD test.

## Data Availability

The data presented in this study are available within the article and Appendix A.

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
