# Peer review of "GhENODL6 Isoforms from the Phytocyanin Gene Family Regulated Verticillium Wilt Resistance in Cotton"

_ijms, 2022, doi:10.3390/ijms23062913_

Round 1

Reviewer 1 Report

I have made a few grammar suggestions directly on the manuscript.  This is an excellent manuscript.  

Author Response

Dear Jaelyn Dong,

We thank you and the reviewers for the constructive comments on our manuscript entitled “GhENODL6 isoforms from the phytocyanin gene family regulate Verticillium wilt resistance in cotton” (ijms-1606084). According to the suggestions, we supplemented the related analysis or experiments including performing the evolutionary tree clustering via the Maximum Likelihood estimation method, detecting the genes (AOS, PDF1.2, JZA1, JZA3, JZA6 and JZA10) expression involved in jasmonic acid pathway in VIGS cotton, and the corresponding results had been added in the revised MS and/or response letter.

Furthermore, we revised the manuscript carefully according to the suggestions. We also invited an English-language editor for critical reading of the revised MS and polishing. Please check the following point-by-point responses (words in red) we have made, and relevant revisions (words in red) in our submitted manuscript.

Best regards.

Yan Zhang & Zhiying Ma

Hebei Agricultural University

[email protected] OR [email protected]

Reviewer 2 Report

In the paper by Zhang et al, the authors identified 41 ENODL genes and subsequently checked them to find the possible role they might play in the resistance of cotton towards wilt pathogen, Verticillium dahliae. The authors showed that the ENODL gene family is divided into 4 subfamilies according to the phylogenetical analysis of the conserved active site. The authors then figured out that one of these genes (GhENODL6) could be the potential target based on the transcriptome analysis. Finally they checked the role of this gene in the resistance of cotton by performing knock out/over expression experiments. The results of this manuscript seems interesting and novel. However, the paper needs to go through a major revision before publication. The writing of the paper is in an acceptable format. However, there are many grammatical and spelling mistakes which could be fixed in the next round of revision. I suggest that the authors ask a native editor to edit their papers before resubmission. The introduction could be improved a bit as it doesn’t cover all aspects of the paper. The discussion of the paper could be improved a bit too as in some parts the authors did not really discuss their findings.  Overall, I believe that the work done by Zhang et al could be publishable in IMA Fungus, but the authors need to put sometimes to do the revisions before resubmission.

Here are some suggestions to improve the quality of this paper:

Line 32: Remove ‘S’ from textiles

Line 35: ‘’There WAS no effective fingicide’’ should be replaced with ’’There IS no effective fingicide’’

Line 36: ‘’WAS approximately 10–35%’’ should be replaced with ‘IS approximately 10–35%’’

Line 38: bring the full name of RMB!

Line 47: You need a reference after ‘’Stress’’ (first word of the sentence)

Line 48: Replace AND with comma.

Line 57: What is CRR1? Is there any full name? What about Chi28?

Line 58: Please complete your sentence. ……..’’in defense response’’ to what? A biotic stress?

Line 69: Add ‘’STUDY’’ after ‘’our previous’’…

Line 83: Replace ‘’PC family member’’ with ‘’PC family members’’

Line 93: you need a reference after ‘’Pfam (PF02298)’’

Line 99: Have you tried other evolutionary methods such as Maximum Likelihood (ML)? If not I suggest to run your data with ML and see if the tree is still same?

Line 102: You can not start your sentence with ‘’Of which’’

Line 113: Again, you can not start your sentence with OF WHICH! Either remove it or put a comma before it and connect it to the previous sentence.

Line 119: After ‘’under V. dahliae stress’’, where are the results? Table…., Figure….??

Line 120: Why did you choose 2 hours specifically?

Line 127: How did you get the information for the protein molecular weight?

Line 132: Replace ‘’he’’ with ‘’THE’’

Line 156: ‘’In additional’’ must be replace with ‘’Additionally’’

Line 158: SYMPTOMS must be replaced with evidences

Line 184: Why did you choose 24hpi in this case?

Line 185: 4127 differentially expressed gene: Previously in line 177 you said the total number of the differentially expressed genes were 4235. Which number is correct?

Line 196: You need to refer to a figure after ‘’qPCR’’.

Line 231: You should replace ‘’In the present,’’ with something like ‘’Up until now’’

Have you also checked the Jasmonic acid pathway response in knocked out plants to see the effect of ENODL6 gene on that pathway? It will be interesting to check that pathway too as V. dahliae is a hemi biotroph pathogen and there are good body of evidences showing the activation of JA pathway in response to such a pathogens.

Line 244: ‘’Plant’’ must be ‘’Plants’’

Line 246: PR gene plays must be PR genes play …..

It will be great to have a small paragraph in introduction and bring some information about different plant defensive systems such as SA and Jasmonic acid pathways etc…

Line 260: pathogenic bacteria! What about fungi? Maybe bring some example of fungi too.

Line 289: ‘’Arabidopsis were grown in a climate’’….. What do you mean by climate? Did you grow them in a green house or a growth chamber? You should clarify it.

Line 360: How did you do the cloning? You should explain the cloning method. Also, where did you get the plasmid? Where can someone find the plasmid map?

Line 378: electric shock method…. Why you didn’t use freeze thaw method as previously? Also you need a reference here.

Line 416: You should explain also how you designed primers? Any softwares??

Which database did you use to do gene ontology analysis? You should bring some more information on gene ontology part.

Figure 2: b and c figures are not clear at all. Please replace them with clearer figures.

Author Response

Dear Jaelyn Dong,

We thank you and the reviewers for the constructive comments on our manuscript entitled “GhENODL6 isoforms from the phytocyanin gene family regulate Verticillium wilt resistance in cotton” (ijms-1606084). According to the suggestions, we supplemented related analysis or experiments including performing the evolutionary tree clustering via the Maximum Likelihood estimation method, detecting the genes (AOS, PDF1.2, JZA1, JZA3, JZA6 and JZA10) expression involved in jasmonic acid pathway in VIGS cotton, and the corresponding results had been added in the revised MS and/or response letter.

Furthermore, we revised the manuscript carefully according to the suggestions. We also invited an English-language editor for critical reading of the revised MS and polishing. Please check the following point-by-point responses (words in red) we have made, and relevant revisions (words in red) in our submitted manuscript.

Best regards.

Yan Zhang & Zhiying Ma

Hebei Agricultural University

[email protected] OR [email protected]

Round 2

Reviewer 2 Report

In general, the authors put some good efforts to answer most of my concerns. The introduction and the results are well explained and the language of the manuscript improved drastically. However, I still have some questions from the authors that need to be addressed in the manuscript before acceptance of this paper.

I wonder why the authors have seen 118 obvious expression of GhENODLs change under V. dahliae stress at 2hpi as shown in section 2.2, but they used 24hpi in section 2.6, when they compared the differentially expressed genes (DEG) between transgenic and 183 WT plants! And they have seen a big expression of the defense genes at 24 hpi! I suggest that they bring some information to explain why that happened in their discussion part.

In terms of gene ontology analysis, please bring the information about the database that you used and not the package! Is it a general database for all plants or is it specific database for cotton?

Author Response

Dear professor Jaelyn Dong,

We thank you and the reviewers for the constructive comments on our manuscript entitled “GhENODL6 isoforms from the phytocyanin gene family regulate Verticillium wilt resistance in cotton” (ijms-1606084). We carefully revised the manuscript according to the Reviewer 2’ suggestions, and the corresponding modification had been added in the revised MS and response letter. Please check the following point-by-point responses (words in red) we have made, and relevant revisions (words in red) in our submitted manuscript.

Best regards.

Yan Zhang & Zhiying Ma

Hebei Agricultural University

[email protected] OR [email protected]
